# Sexual Orientation Microaggression Experiences and Coping Responses of Lesbian, Gay, and Bisexual Individuals in Taiwan: A Qualitative Study

**DOI:** 10.3390/ijerph20032304

**Published:** 2023-01-28

**Authors:** Yu-Te Huang, Wen-Jiun Chou, Yi-Chen Hang, Cheng-Fang Yen

**Affiliations:** 1Department of Social Work and Social Administration, The University of Hong Kong, Hong Kong RM543, China; 2Department of Child and Adolescent Psychiatry, Chang Gung Memorial Hospital, Kaohsiung Medical Center, Kaohsiung 83301, Taiwan; 3School of Medicine, Chang Gung University, Taoyuan 33302, Taiwan; 4Taiwan Gender Equity Education Association, Taipei 10089, Taiwan; 5Department of Psychiatry, Kaohsiung Medical University Hospital, Kaohsiung 80708, Taiwan; 6Department of Psychiatry, School of Medicine, Kaohsiung Medical University, Kaohsiung 80756, Taiwan; 7College of Professional Studies, National Pingtung University of Science and Technology, Pingtung 91201, Taiwan

**Keywords:** microaggression, sexual orientation, lesbian, gay, bisexual, mental health, qualitative study

## Abstract

This qualitative study explored the sexual orientation microaggression (SOM) experiences and coping strategies of lesbian, gay, and bisexual (LGB) individuals in Taiwan. In total, 30 LGB individuals (17 women and 13 men; 17 homosexual and 13 bisexual individuals) who experienced SOMs participated in qualitative, semistructured interviews, during which their SOM experiences were assessed. Through the interviews, several types of SOMs were identified, corresponding to three main types of microaggression (microassaults, microinsults, and microinvalidations) previously identified in Western studies. The participants reported various coping responses to SOMs, ranging from active responses to choosing not to respond, to protecting themselves or to minimizing the negative consequences of confrontation. The results provide mental health professionals with insight regarding the contexts of and coping responses to the SOMs experienced by LGB individuals.

## 1. Introduction

Sexual orientation microaggressions (SOM) are a form of sexual stigmatization that is commonly experienced by lesbian, gay, and bisexual (LGB) individuals [1,2]. Sue et al. asserted that microaggression involves brief indignity and manifests as verbal expressions, behaviors, or environmental planning that affects individuals with minority characteristics (e.g., ethnic and sexual minorities) [3]. SOMs, intentional or not, are rooted in heterosexism and homophobia and are often hostile, derogatory insults directed at LGB individuals [3,4]. Studies have identified three forms of SOMs, namely microassaults (i.e., discriminatory behaviors against individuals with sexual minority identities), microinsults (i.e., subtle slights against individuals with sexual minority identities), and microinvalidations (i.e., nullification of the stigmatization-related experiences of individuals with sexual minority identities) [4,5,6].

The importance of exploring LGB individuals’ SOM experiences and coping responses can be elaborated via the framework of minority stress theory [7] and the psychological mediation framework of stigma [8]. According to minority stress theory [7], SOMs and coping mechanisms are distal and proximal stressors, respectively; both can increase LGB individuals’ distress and the risks of mental health problems [5,9]. For example, LGB individuals may cope with prejudice events by concealing their sexual orientation; concealment of sexual orientation can also distress LGB individuals and relate to affective symptoms and poor mental well-being [10]. According to the psychological mediation framework of stigma [8], coping is one of general psychological process that mediates the relationship between SOMs and mental health problems. Therefore, increasing the knowledge of LGB individuals’ SOM and coping responses to SOM is important to the development of intervention strategies for health promotion.

SOMs are less frequently reported by victims and perpetrators than are the overt stigmatizing attitudes and behaviors [4,5]. Moreover, most studies of SOMs have been conducted in the North America [11,12,13,14,15]. Relative to people in Western societies, those in East Asian societies report less favorable attitudes toward LGB individuals [16]. For example, societies (e.g., Taiwan, Hong Kong, and China) that are influenced by traditional Confucianism tend to endorse the belief that people need to form families and have children carry on their family bloodline and fulfill their responsibility of filial piety [17]. Studies have reported that sexuality-related bullying and the related mental health problems are prevalent among LGB individuals in Taiwan [18,19,20,21,22,23,24,25,26,27]. Moreover, according to the psychological mediation framework of stigma [8], sociocultural backgrounds may moderate the relationship between SOMs and coping responses in LGB individuals. According to ecological system theory, mental health is the result of interactions between the individuals and multiple systems, including the microsystem, mesosystem, exosystem, and macrosystem [28]. SOMs are the result of the stigmatizing treatment of multiple systems on LGB individuals; LGB individuals’ coping responses to SOM may also influence themselves and the systems. Thus, the SOM experiences and coping responses of LGB individuals in Taiwan warrant academic and clinical attention.

Because of their subtle and covert characteristics, SOMs are frequently overlooked and trivialized by the public and even LGB individuals themselves. Therefore, qualitative studies of what LGB individuals experience and how they cope with SOMs can provide a key reference for developing intervention programs. Various experiences of SOMs have been identified in previous studies in various environments, such as in the community [6,29,30], on campus [11,13,17], in psychotherapeutic environments [12,14], and in the family [31]. Common manifestations of SOM include stereotypical assumptions (e.g., presenting stereotypical beliefs about LGB individuals regarding appearances, sexual activities, and romantic relationships) [6,12,13,29,31], homophobia (e.g., assuming that LGB orientation is contagious and that LGB individuals should be avoided) [6,13,14,29,31], heterosexist language or terminology (e.g., using language reflecting a heteronormative value system that derogates LGB individuals) [6,12,13,17,29,30], the sinfulness or criminality (e.g., believing that LGB orientation is morally deviant) [6,11,14,29,30], the assumption of psychological abnormality (e.g., assuming that LGB individuals need therapy) [6,12,14,17], invalidation of experience (e.g., questioning the reality or importance of LGB individuals’ feeling and experiences) [14,30,31], the denial of individual heterosexism (e.g., heterosexual individuals deny any biases or unfavorable attitudes toward LGB individuals they may hold) [6], the endorsement of heteronormative culture and behaviors (e.g., recognizing that the standards of heterosexual individuals are the only norm in the world) [6,12,13,14,17,29,30], undersexualization (e.g., accepting sexual orientation of LGB individuals in a surface level) [12,29], microaggressions as humor (e.g., delivering microaggression in a joking manner) [29], and the allowing of institutionally endorsed microaggressions [14,17]. However, very few qualitative studies have been conducted outside the North America. A qualitative study explored the SOM experiences of LGB students aged 16 or 17 years in Hong Kong [17]. Because of the low level of acceptance of LGB individuals in Asian societies [16], whether the manifestations of SOMs identified in previous studies also exist in Taiwan needs further study.

LGB individuals may adopt various coping strategies in response to overt sexuality-related prejudice and discrimination, such as concealing their sexual orientation [31,32,33,34], seeking support [34], and challenging heterosexist assumptions [34]. For example, a quantitative study of men who have sex with men in China found that adopting avoidant coping methods to deal with sexual stigma predicted depression and anxiety [35]. A qualitative study on gay and bisexual young men in the United States found a range of strategies that gay and bisexual youth use to protect themselves from the detrimental effects of heterosexism, including situation selection, situation modification, attentional deployment, cognitive change, and response modulation [36]. However, compared with the studies on the coping responses of LGB individuals to overt stigmatization, very few qualitative studies examined the coping responses of LGB individuals to SOMs. A qualitative study on young gay and bisexual Latino men identified three strategies of resilience used to endure or overcome SOMs from family members, namely, self-discovery (e.g., seeking information and engaging in community mobilization), adaptive socialization (e.g., orienting oneself to thrive socially by being aware of but not consumed by hostile influences), and self-advocacy (empowering oneself to represent one’s values) [31]. Understanding the contexts of SOMs and the coping responses to them is helpful for developing intervention strategies that LGB individuals can apply to cope with SOM experiences and maintain their mental health [31].

This qualitative study explored the SOM experiences of LGB individuals in Taiwan and the corresponding coping strategies. Because a comprehensive and contextualized understanding of stigmatization-related experiences is fundamental to the development of interventions [37], we employed a phenomenological approach. According to minority stress theory [7] and the psychological mediation framework of stigma [8], LGB individuals experience SOMs and SOM-induced stress in their social environments; therefore, two research questions were formulated for the present study: (1) What are the types of SOMs experienced by LGB Taiwanese individuals in their interactions with their family, peers, colleagues, and others, and (2) what are the psychosocial and behavioral strategies adopted by these LGB individuals to manage SOMs?

## 2. Materials and Methods

### 2.1. Participants and Procedure

The present study recruited its participants from among the individuals who participated in the Taiwanese Study of Sexual Stigma (T-SSS) [38]. The participant inclusion criteria of the T-SSS were Taiwanese individuals who identified their sexual orientation as being gay/lesbian or bisexual, aged between 20 and 30 years. Participants were recruited by posting an online advertisement on social media including *Facebook*, *Twitter*, *LINE* (a direct messaging app), and Bulletin Board System (a popular application dedicated to sharing or exchange of messages on a network) from August 2018 to July 2020. In total, 1000 participants (500 males and 500 females) participated in the study. Multiple forms of sexual stigma, including perceived sexual stigma from family and friends, internalized sexual stigma, and SOMs were collected by self-reported questionnaires. The SOM experiences were assessed using the traditional Chinese version [38] of the Sexual Orientation Microaggression Inventory (SOMI) [3]. The participants rated the 19 items of the SOMI on a 5-point scale, with endpoints ranging from 1 (not at all) to 5 (almost every day); therefore, the total SOMI scores ranged from 19 to 95, with a higher total score indicating more SOM experiences. The present study sent the invitation message by cell phones to the 50 LGB individuals who obtained the highest SOMI scores on the T-SSS to invite them to participate in qualitative interviews. In total, 30 LGB individuals (17 women and 13 men) responded to the invitation and agreed to participate. We estimated this sample size to be enough according to Guest et al.’s suggestion when we composed a funding proposal for this study, and interview data from this cohort of participants appeared adequate to answer the research questions [39]. Among them, 27 participants had a bachelor’s degree, and 3 had a high school diploma; 17 participants self-identified as gay or lesbian, and 13 self-identified as bisexual; 8 participants came from north Taiwan, 11 from central Taiwan, and 11 from southern Taiwan.

The participants participated in in-depth interviews between August 2020 and January 2021. Informed consent was obtained from all participants prior to the interviews. The present study was approved by the Institutional Review Board of Kaohsiung Medical University Hospital (KMUHIRB-F(II)-20180018).

### 2.2. Data Collection

One researcher (CFY) conducted a semistructured interview in Mandarin with each participant individually in an interview room. The interviewer elicited information about SOM experiences from the participants by framing our questioning as follows: “You participated in our previous survey and reported that some things that people did or said made you feel uncomfortable or indignant because you are gay, lesbian or bisexual. Could you tell me what happened? What are some subtle ways that people treat you differently or negatively because of your sexual orientation? Have you heard or seen someone do or say anything that appeared to reflect prejudice against gay, lesbian, or bisexual people but was not directed at you? Tell me what you heard or saw.” To explore the participants’ coping responses to SOMs, we asked them questions such as “How did you respond to that?” and “What made you decide to handle things that way?” Each interview lasted between 45 and 60 min. All interviews were recorded and transcribed verbatim in their original language, and selected quotations were translated into English for presentational purposes. 

### 2.3. Data Analysis

Two researchers (YTH and YCH) jointly performed a thematic analysis with the aid of NVivo 10 software (Lumivero, Denver, CO, USA) to analyze the transcripts [40]. Content analysis was not applied since the study objective was to offer comprehensive descriptions of SOM experiences rather than to determine the prevalence or intensity of a certain form of SOMs. At the first stage, two coders read the transcripts thoroughly to familiarize themselves with the data. Stage two involved initial line-by-line coding, whereby descriptive codes were assigned to indicate the meanings in the text. At stage three, the analysts conducted focused coding to identify themes from the codes according to their nature and relevance to experiences and coping methods of microaggression. Because the study objective was to identify the forms and patterns of SOMs that the participants experienced which targeted their LGB identities, the researchers differentiated SOM experiences from overt forms of discrimination by reviewing the interview transcripts. A framework was created on the basis of the codes generated from the analyses of the first few transcripts. Subsequently, themes related to the research questions were identified and conceptualized. Finally, we defined the meanings of themes and assessed whether they adequately answered the research questions. The coded themes were then reviewed through regular research team discussion to assess whether the themes had been consistently and properly developed.

## 3. Results

The interviews revealed numerous incidents and experiences characterized by subtle, nuanced forms of SOMs. After reviewing the characteristics and effects of these SOM experiences, we classified them as discriminatory remarks (a type of microassault), sexuality-related stereotyping (a type of microinsult), or the microinvalidation of same-sex relationships. The interviewees also described their coping responses to these SOMs.

### 3.1. Discriminatory Remarks

In the present study, 26 participants reported notable incidents that categorized as involving discriminatory remarks, which constitute a type of microassault. The recurring messages that characterized these discriminatory remarks included that gay men are girly/feminine, that many lesbians are manly/masculine, that LGB people are not normal, and that such individuals making these remarks stay away from LGB individuals to avoid feeling uneasy. Participant #10 (a lesbian woman) described the following example of verbal assault on bisexual people:

Some people are very mean, and they like to say “Yuck! You can do both men and women” followed by “Do not you think that is strange?” What ordinary people say to homosexual people they also say to bisexual people.

Another lesbian woman (Participant #26) recalled the following experience:

I think it happened in junior high school, when we were all immature. So, our classmates would use this as a topic of conversation and talk behind other students, saying “He’s so gay! Not a normal man!” Because they were only junior high school students and were immature, they would gossip for fun.

Four participants reported frequent experiences of social isolation due to discriminatory remarks. Participant #17 (a lesbian woman) recalled how people avoided interacting with her because they were unsure of her sexual orientation.

I used to wear gender-neutral clothing and wore my hair short since I was a student. My classmates suspected if I liked women, but they did not ask me directly and just avoided interacting with me. … Of course, when my hair grew back again later, I felt the difference in how society saw me.

Because hostile and rejecting attitudes often became noticeable, numerous participants felt the need to be cautious about how they responded (i.e., their coping responses) to these discriminatory remarks. In several scenarios, the participants responded actively and intentionally. In response to Participant #10 (a bisexual woman) recounted the following: “I would always tell the person, ‘You do not have to like it, but you do not need to criticize others.’” She was then asked, “Do you always say that when you hear such things, or do you choose who to speak to?” She responded: “No, I speak regardless of the situation because I think it’s a matter of personal choice and personal feelings. Why are they concerned about what others like to do? Why do they have to care so much?”

Participant #12 (a gay man) adopted the strategy of selectively communicating with people who might be ready to discuss issues relating to sexual and gender identity:

Some people, such as family members, do not understand sexual and identity, so I will not discuss this with them. I will discuss this issue only with people who understand, which saves time because what everyone knows is different. … It’s only because they do not understand this aspect of my life; if it’s a conversation about other matters such as family life, casual topics, or other things, we can still chat. I do not necessarily have to focus on this matter. … We can go out and have fun or do whatever they want, but I will be aware that I cannot talk about gender issues with this person.

Other participants emphasized that their decision to adopt a confrontational attitude is dependent on the characteristics of the people they interact with. An attempt to communicate with an individual could be unsuccessful if that individual is not open minded or exhibits an indifferent attitude. Participant #16 (a bisexual woman) elaborated on this experience:

It depends on the type of person with whom I interact. If they are open to new ideas, I will try to hang around and be the angel on their shoulder (i.e., give them advice). … If they are politically indifferent, they may be people who make decisions on the basis of their limited knowledge, so, I would feel they can learn more because it already means something has gotten their attention. Then, I would throw a different set of ideas at them, saying “Well, actually, you can take a look; things are not necessarily the way you think they are.”

By contrast, some participants turned to professional help to cope with microassault experiences. Participant #11 (a gay man) recalled the following experience:

I used to go to the counseling room frequently when I was in high school. The counselor was gay, but because this aspect of his identity was concealed, he would talk about how I can resolve the situation. So, the counseling room was also support for me back then.

Several participants noted that, to protect themselves or minimize the negative consequences of confrontational responses, they would not respond to discriminatory remarks. Participant #11 (a gay man) shared his reason for not responding to microassaults in specific situations:

Because I do not really care about others’ opinions, I managed to stumble along all this time. If I really did care about their opinions, then I may not be alive today. … So if I were to have taken their words to heart, then I would not have been able to survive then.

He continued:

Because if you get deeply hurt, you will not have the strength to face your life, so you protect yourself and your life as a priority. Once your life is stable and you have the capacity, you can then assess whether you want to educate others or recharge yourself. Protect yourself first; do not let yourself be knocked down first. That is the most important thing.

### 3.2. Sexuality-Related Stereotypes

Sixteen participants recalled experiences of being stereotyped by others; such experiences are similar to microinsults. These stereotypical remarks were often made by people who did not appear hostile or aggressive in their intentions or demeanor; therefore, they were not necessarily perceived as overt insults or threats. However, the stereotyping still aroused negative emotions in the participants because it betrayed ingrained prejudices regarding the psychological and behavioral characteristics of sexual minority individuals. A common type of stereotypical remark that upset many of the participants was the linking the LGB identity with an increased risk of human HIV infection and AIDS. Participant #8 (a lesbian woman) shared the following:

My classmates’ attitude is more like “because you are my friend or someone I know, I think you are a good person; but that does not mean that homosexuality is good.” I think there will still be people who think that being gay is a relatively bad phenomenon and that they are “prone to kill their lovers, engage in orgies, and be infected with HIV and so on, but not you.”

This prejudicial statement was not directed at the participant; thus, it was not viewed as a verbal insult or humiliating statement. However, the misconception about the link between having an LGB identity and contracting specific diseases was regarded by this participant as ignorant and offensive. Similarly, Participant #12 (a gay man) shared his experience of hearing prejudicial comments from his family:

It is common for one’s family or relatively close friends to make such remarks. When they did not know about my sexual orientation, they actually made several offensive jokes such as that gay people must have AIDS or that they are promiscuous or something.

Another common stereotype involves the imposition of heterosexism and clear-cut gender roles onto intimate relationships. Fundamentally, this microinsult represents a dismissal of the complexity and uniqueness of individual romantic relationships. Participant #25 (a bisexual woman) described the following incident:

My thesis advisor kept asking me, “Which one of you is the man and which one is the woman?” He thought that because I had a boyfriend, my preferred must be gender-neutral people or even people who play the masculine role.

Another lesbian participant (Participant #4) was routinely asked similar questions about her same-sex relationship:

Regarding this matter, you will always be asked, “So are you the man or the woman?” In most common conversations, I reply with “I am a woman,“ and they follow up with “So your girlfriend is the man?” to which I say, “No, she is also a woman.”

In addition to the aforementioned examples, the participants gave examples of various stereotypes about LGB individuals, including that LGB individuals always find other same-sex people attractive (three participants), LGB individuals are promiscuous (five participants) and exhibitionist (one participant), most gay men are good looking (two participants), LGB individuals must have had bad opposite-sex relationships (one participant), lesbians who exhibit a masculine identity are self-righteous (one participant), crimes of passion are common among same-sex couples (two participants), and childhood trauma affects sexual orientation (two participants).

To respond to remarks about these stereotypes, which are often subtle and indirect, the participants adopted various strategies depending on the identity of the individual making the remark and the context of the remark. Active responses included asking for clarification (three participants), engaging in discussion with a trusting person (one participant), responding selectively (four participants), occasionally disputing statements (two participants), engaging in rational discussion (two participants), and teasing in response (one participant). Participant #2 (a gay man) described his experience as follows:

I am anxious about straight men having inaccurate stereotypes about gay men, such as “gays like all men.” I make it clear from the beginning that “I can be friends with you and I am gay, but I am not interested in you; we can be just friends.”

Participant #19 (a bisexual woman) described her response to stereotypical comments:

If it is not an in-person interaction or is simply an online interaction on the Internet, YouTube, Instagram, Facebook, or some other forum, I may leave a message below and reply rationally to tell the person that they are stereotyping.

Several participants indicated that they tended not to respond to stereotypical remarks, because they had no opportunities to provide an explanation (one participant); instead, they remained silent (three people), evaded or changed the topic (one participant), pretended to be straight (three participants), or ignored such remarks (one participant). Participant #24 said the following:

Although I would be a little uncomfortable or would want to refute this remark to some extent, I do not want to let them know the fact that I am [gay] or make them become suspicious of me, so I will not say anything about it or discuss this matter with them.

### 3.3. Invalidation of Same-Sex Relationships

Five participants described microinvalidation experiences in which their sexual identities were dismissed and invalidated. A prominent theme of this type of microinvalidation was the transitional nature and reversibility of same-sex attraction. A lesbian participant (Participant #5) provided the following example:

I have admitted to my dad that I like women and that I have a girlfriend, but my dad felt that … well … he did not outright say whether this was good or bad; he only said that this is just a phase, that my mother once felt affection toward women too, and that this will change in the future.

Four participants noted a distinct type of invalidation in which their sexual identities were nullified and described as a “trend”. Specifically, some people attributed the participants’ sexual orientation to their personal intention to follow new, popular trends; that is, they regarded the participants’ sexual orientation as a modifiable and deliberate choice. Participant #17 (a lesbian woman) described her experience as follows:

I feel I simply like women, but when I am with her, people might say “lesbianism is all the fad these days.” They would use the word *fad*, which makes me uncomfortable. I feel that I just simply like women, but some people would remark that being homosexual is only temporary, which is very uncomfortable to hear.

Participant #14 (a lesbian woman) shared a similar experience:

I tell them that I have a girlfriend, and some would then say “Is this very popular now? Why? Is this a fad? Why would a good girl like you end up with other girls?”

In particular, questions about nonheterosexual orientations are frequently directed at bisexual individuals. Participant #20 (a bisexual woman) explained this prevalent type of microinvalidation experienced by bisexual people:

People may think that bisexual people have a choice and that bisexual people are just temporarily confused. … They say that bisexual women may have been simply attracted to handsome women and that is why they became a couple. However, basically, they like handsome-looking people, so they are actually straight.

## 4. Discussion

The present study on LGB individuals in Taiwan identified several types of SOMs that correspond to three types of microaggressions (i.e., microassaults, microinsults, and microinvalidations) previously identified in several studies conducted in Western countries [3,4,5,6]. Furthermore, the present study identified various coping responses to SOMs adopted by LGB individuals in Taiwan, ranging from active responses to choosing not to respond.

First, this study identified microassaults as a type of microaggression that manifests as discriminatory remarks regarding LGB identities. The interviewed LGB individuals experienced microassaults resulting from the expectation of some people that everyone should exhibit gender-conforming behavioral and psychological traits [41] regardless of their sexual orientation. Furthermore, some people exclusively endorse heteronormality and consider LGB individuals to be abnormal. The findings of this study echoed the SOM experiences identified in previous studies conducted in the North America and Hong Kong such as endorsement of heteronormative culture and behaviors, assumption of psychological abnormality, and avoiding contacts due to homophobia [6,12,13,14,17,29,30,31]. The present study revealed that LGB individuals experienced microassaults from early adolescence onwards. In addition to overt and personal criticisms, social isolation due to these discriminatory remarks often affects LGB individuals. Researchers have discovered that discriminatory remarks and the resulting social isolation have a harmful effect on the mental health of young LGB individuals [26,27,42]. The present study revealed that the interviewed LGB individuals found it challenging to cope with microassaults. Although several participants actively and intentionally responded to microassaults, many of them selected their strategies on the basis of the situation and the identity of the individual who perpetrated the microassault. For example, several participants discussed their sexual orientation only with people who were open to discussing stereotyping problems. The participants also indicated that to protect themselves from the unpredictable consequences of responses, they chose to not confront individuals making these discriminatory remarks. These results suggest that, in situations involving microassaults, LGB individuals have little power and find it difficult to challenge discriminatory remarks by themselves.

The present study revealed that sexuality stereotyping is a common type of microinsult. Similar SOM manifested with stereotypical assumptions were also identified in previous studies conducted in the North America [6,12,13,29,31]. People often link the LGB identity with promiscuity, HIV infection, and AIDS. These sexuality-related stereotypes have existed for such a long time that they are regarded as true by many people. The notion of clear-cut gender roles that conform to heterosexual relationships is another common sexuality-related stereotype. This SOM rooted in the recognition that the standards of heterosexual individuals are the only norm in the world [6,12,13,14,17,29,30]. LGB individuals may face impolite inquiries from heterosexist acquaintances or strangers. However, the perpetrators of microinsults might underestimate the harm that they cause to LGB individuals. Similarly, heterosexual individuals often deny any biases or unfavorable attitudes toward LGB individuals they may hold [6]. Moreover, the victims of microinsults may find it difficult to communicate to the perpetuators of stereotypes about their reality and how they feel about such microinsults [43]. Several participants indicated that they actively respond to sexuality-related stereotyping, whereas other adopt various types of coping responses depending on who perpetrates the microinsult. A reason for adopting a passive response or nonresponse is the concern about disclosing one’s sexual orientation; this indicates that an individual’s coping response to SOMs is deeply influenced by expectations regarding the negative consequences of sexual orientation disclosure.

The present study revealed that people often perceive same-sex relationships as a trend and invalidate their existence. Previous studies have also revealed that questioning the reality or importance of LGB individuals’ feeling and experiences is one of the common SOMs in western societies [14,30,31]. Societal tolerance of LGB identities in Taiwan has increased over the past 2 decades [16]. Numerous LGB individuals are now willing to disclose their sexual orientation and same-sex partnerships. However, numerous people in Taiwan still exhibit homonegative attitudes and invalidate the same-sex relationships of LGB individuals. On 24 November 2018, the people of Taiwan voted in a referendum to oppose the amendment of the Civil Code to legalize same-sex marriage [44,45]. The result of this referendum indicates that microinvalidation of same-sex relationships may persist, alongside direct and overt invalidation behaviors.

The results of this qualitative study suggest that interventions at the individual and environmental levels are required to mitigate SOMs. On the individual level, LGB individuals are still in a low-power position with respect to situations in which addressing SOMs and confronting the perpetrators of SOMs are options. Empowerment-oriented interventions and campaigns are required to enable LGB individuals to develop alternative strategies for coping with SOM experiences. At the systemic and institutional levels, governments can play a proactive role in establishing antidiscrimination laws and policies to reduce discriminatory remarks, sexuality-related stereotyping, and the microinvalidation of same-sex relationships [46].

### Limitations

This study has several limitations. First, the study sample comprised young LGB adults who reported extensive SOM experiences on the SOMI. Whether the results of the present study can be generalized to other age groups or LGB individuals without severe SOM experiences is unclear. Second, the study participants’ accounts regarding their responses to microinvalidation were limited. Third, the present study did not distinguish the SOM experiences of the LGB individuals by specific sexual orientation. Future studies should collect more data on coping responses to microinvalidation and examine the SOM experiences of various subgroups of LGB individuals.

## 5. Conclusions

Similar to the study by Sue et al. [3], the present qualitative study identified three types of SOM experienced by LGB individuals in Taiwan. The interviewed LGB individuals adopted various coping responses to SOMs, and they were concerned about disclosure and the safety-related problems associated with active responses to SOMs. On the individual level, the present study discovered that LGB individuals did not always have the power to openly communicate with the perpetrators of SOMs. LGB individuals must be empowered to develop alternative strategies for coping with SOM experiences. Individual and systemic interventions are warranted to reduce the prevalence of SOMs and assist LGB individuals in coping with SOMs.

## Data Availability

The qualitative data will be unavailable to safeguard participants’ privacy.

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
