# Peer review of "Sexual Orientation Microaggression Experiences and Coping Responses of Lesbian, Gay, and Bisexual Individuals in Taiwan: A Qualitative Study"

_ijerph, 2023, doi:10.3390/ijerph20032304_

Round 1

Reviewer 1 Report

The article has potential, but it requires some degree of revision before it can be considered for publication.

First, there must be a dedicated section for the literature review. More needs to be known about the phenomenon not only in the context in which the study was conducted, but also elsewhere given the fact that SOMS experiences and coping strategies form an important aspect of the lives of LGBTQ the world over. The literature review should also help establish what has been done so far in regard to research and most importantly, the gap(s) that the authors sought to address in/through their study. 

Second, there must be a theoretical framework that provides a specific perspective or lens through which the phenomenon can be examined more meaningfully. While the minority stress theory and the psychological medication framework are mentioned, these should be explained further in accordance with the aim and the objectives of the entire study - which brings up the question of whether the conceptual framework ought to be included as well.

Third, the data collection procedures, particularly the recruitment of interview respondents require further information: How were they ‘actually’ recruited? (e.g., personal contacts, snowballing? Even the T-SSS requires some details!) Which parts of Taiwan are they from? Were non-Taiwanese included? Did anyone decline the invitation to the interview and why? The data analysis procedures, especially the methods used to analyze the interview transcripts need further elaboration: was the qualitative content analysis approach used? Or was it a thematic approach (e.g., Braun and Clarke's TA)? - which brings up another question on the research design for the whole study: Case study? Phenomenology?

Fourth, the discussion of findings should be revised once the literature review section is added to the article. 

Thank you to the authors and all the best with the revision!

Author Response

We appreciated your valuable comments. As discussed below, we have revised our manuscript with underlines based on your suggestions. Please let us know if we need to provide anything else regarding this revision.

Comment 1

First, there must be a dedicated section for the literature review. More needs to be known about the phenomenon not only in the context in which the study was conducted, but also elsewhere given the fact that SOMS experiences and coping strategies form an important aspect of the lives of LGBTQ the world over. The literature review should also help establish what has been done so far in regard to research and most importantly, the gap(s) that the authors sought to address in/through their study. 

Response

Thank you for your comment. We rewrote the literature review and added more introduction for SOM experiences and coping strategies as below. Please refer to 56-64 and 72-117.

SOMs are less frequently reported by victims and perpetrators than are the overt stigmatizing attitudes and behaviors [4,5]. Moreover, most studies of SOMs have been conducted in the North America [11–15]. Relative to people in Western societies, those in East Asian societies report less favorable attitudes toward LGB individuals [16]. For example, societies (e.g., Taiwan, Hong Kong, and China) that are influenced by traditional Confucianism tend to endorse the belief that people need to form families and have children carry on their family bloodline and fulfill their responsibility of filial piety [17]. Studies have reported that sexuality-related bullying and the related mental health problems are prevalent among LGB individuals in Taiwan [18–27].

Various experiences of SOM have been identified in previous studies in various environments, such as in community [6,29,30], campus [11,13,17], psychotherapeutic environments [12,14], and family [31]. Common manifestations of SOM include stereotypical assumptions (e.g., presenting stereotypical beliefs about LGB individuals regarding appearances, sexual activities, and romantic relationships) [6,12,13,29,31], homophobia (e.g., assuming that LGB orientation is contagious and that LGB individuals should be avoided) [6,13,14,29,31], heterosexist language or terminology (e.g., using language reflecting a heteronormative value system that derogate LGB individuals) [6,12,13,17,29,30], sinfulness or criminality (e.g., believing that LGB orientation is morally deviant) [6,11,14,29,30], assumption of psychological abnormality (e.g., assuming that LGB individuals need therapy) [6,12,14,17], invalidation of experience (e.g., questioning the reality or importance of LGB individuals’ feeling and experiences) [14,30,31], denial of individual heterosexism (e.g., heterosexual individuals deny any biases or unfavorable attitudes toward LGB individuals they may hold) [6], endorsement of heteronormative culture and behaviors (e.g., recognizing that the standards of heterosexual individuals are the only norm in the world) [6,1214,17,29,30], undersexualization (e.g., accepting sexual orientation of LGB individuals in a surface level) [12,29], microaggressions as humor (e.g., delivering microaggression in a joking manner) [29], and allowing of institutionally endorsed microaggressions [14,17]. However, very few qualitative studies were conducted outside the North America. A qualitative study explored the SOM experiences of LGB students aged 16 or 17 years in Hong Kong [17]. Because of low acceptance of LGB individuals in Asian societies [16], whether the manifestations of SOM identified in previous studies also exist in Taiwan needs further study.

LGB individuals may adopt various coping strategies in response to overt sexuality-related prejudice and discrimination, such as concealing their sexual orientation [31–34], seeking support [34], and challenging heterosexist assumptions [34]. For example, a quantitative study on men who have sex with men in China found that adopting avoidant coping with sexual stigma predicted depression and anxiety [35]. A qualitative study on gay and bisexual young men in the United States found a range of strategies that gay and bisexual youth use to protect themselves from the detrimental effects of heterosexism, including situation selection, situation modification, attentional deployment, cognitive change, and response modulation [36]. However, compared with the studies on the coping responses of LGB individuals to overt stigmatization, very few qualitative studies examined the coping responses of LGB individuals to SOMs. A qualitative study on young gay and bisexual Latino men identified three strategies of resilience used to endure or overcome SOMs from family members, namely, self-discovery (e.g., seeking information and engaging in community mobilization), adaptive socialization (e.g., orienting oneself to thrive socially by being aware of but not consumed by hostile influences), and self-advocacy (empowering oneself to represent one’s values) [31]. Understanding the contexts of SOMs and the coping responses to them is helpful for developing intervention strategies that LGB individuals can apply to cope with SOM experiences and maintain their mental health [31].

Comment 2

Second, there must be a theoretical framework that provides a specific perspective or lens through which the phenomenon can be examined more meaningfully. While the minority stress theory and the psychological medication framework are mentioned, these should be explained further in accordance with the aim and the objectives of the entire study - which brings up the question of whether the conceptual framework ought to be included as well.

Response

We added more introduction regarding the conceptual framework according to minority stress theory, psychological medication framework, and ecological system theory into Introduction section as below.

According to minority stress theory [7], SOM and coping mechanisms are distal and proximal stressors, respectively; both can increase LGB individuals’ distress and the risks of mental health problems [5,9]. For example, LGB individuals may cope with prejudice events by concealing their sexual orientation; concealment of sexual orientation can also distress LGB individuals and relate to affective symptoms and poor mental well-being [10]. According to psychological mediation framework of stigma [8], coping is one of general psychological process that mediate the relationship between SOM and mental health problems. Therefore, increasing knowledge of LGB individuals’ SOM and coping responses to SOM is important to the development of intervention strategies for health promotion.” Please refer to 46-55.

“Moreover, according to psychological mediation framework of stigma [8], sociocultural backgrounds may moderate the relationship between SOM and coping responses in LGB individuals. According to ecological system theory, mental health is the result of interactions between the individuals and multiple systems, including microsystem, mesosystem, exosystem, and macrosystem [28]. SOM is the stigmatizing treatment of multiple systems on LGB individuals; LGB individuals’ coping responses to SOM may also influence themselves and the systems. Thus, the SOM experiences and coping responses of LGB individuals in Taiwan warrant academic and clinical attention.” Please refer to 64-72.

Comment 3-1

The data collection procedures, particularly the recruitment of interview respondents require further information: How were they ‘actually’ recruited? (e.g., personal contacts, snowballing? The T-SSS requires some details.)

Response

We added introduction regarding the data collection procedures as below. Please refer to line 130-144.

The present study recruited its participants from among the individuals who participated in the Taiwanese Study of Sexual Stigma (T-SSS) [38]. The participant inclusion criteria of the T-SSS were Taiwanese individuals who identified their sexual orientation as being gay/lesbian or bisexual, aged between 20 and 30 years. Participants were recruited by posting an online advertisement on social media including Facebook, Twitter, and LINE (a direct messaging app), and Bulletin Board System (a popular application dedicated to sharing or exchange of messages on a network) from August 2018 to July 2020. In total, 1000 participants (500 males and 500 females) participated in the study. Multiple forms of sexual stigma, including perceived sexual stigma from family and friends, internalized sexual stigma, and SOM were collected by self-reported questionnaires. The SOM experiences were assessed using the traditional Chinese version [38] of the Sexual Orientation Microaggression Inventory (SOMI) [3]. The participants rated the 19 items of the SOMI on a 5-point scale with endpoints ranging from 1 (not at all) to 5 (almost every day); therefore, the total SOMI scores ranged from 19 to 95, with a higher total score indicating more SOM experiences.”

Comment 3-2

Which parts of Taiwan are they from?

Response

We added the explanation as below. Please refer to line 152-153.

.8 participants came from north Taiwan, 11 from middle Taiwan, and 11 from southern Taiwan.

Comment 3-3

Were non-Taiwanese included?

Response

This study did not include non-Taiwanese. Please refer to line 132.

The participant inclusion criteria of the T-SSS were Taiwanese individuals..

Comment 3-4

Did anyone decline the invitation to the interview and why?

Response

We sent the invitation message to 50 LGB individuals; 30 invited individuals responded to the message and participate into this study. Please refer to line 144-147.

“The present study sent the invitation message by cell phones to the 50 LGB individuals who obtained the highest SOMI scores on the T-SSS to participate in qualitative interviews. In total, 30 LGB individuals (17 women and 13 men) responded to the invitation and agreed to participate.

Comment 3-5

The data analysis procedures, especially the methods used to analyze the interview transcripts need further elaboration: was the qualitative content analysis approach used? Or was it a thematic approach (e.g., Braun and Clarke's TA)? - which brings up another question on the research design for the whole study: Case study? Phenomenology?

Response

We elaborate on how we performed the thematic analysis to analyze the data. As a part of a larger mixed-methods study, the data for this qualitative study were collected by asking the participants to talk about their previous experiences and coping with various forms of microaggression. Instead of adopting a specific methodology to explore these experiences, thematic analysis was followed to guide the data analysis process.

Two researchers (YTH and YCH) jointly performed a thematic analysis with the aid of NVivo 10 software to analyze the transcripts [40]. Content analysis was not applied since the study objective was to offer comprehensive descriptions of SOM experiences rather than to determine the prevalence or intensity of a certain form of SOMs. At the first stage, two coders read the transcripts thoroughly to familiarize themselves with the data. Stage Two involved initial line-by-line coding, where descriptive codes were assigned to indicate the meanings in the text. At Stage Three, the analysts conducted focused coding to identify themes from the codes according to their nature and relevance with experiences and coping methods of microaggression.” Please refer to line 174-182.

The coded themes were then reviewed through regular research team discussion to assess whether the themes had been consistently and properly developed.” Please refer to line 189-190.

Comment 4

Fourth, the discussion of findings should be revised once the literature review section is added to the article. 

Response

We revised the Discussion section and added the contents below into the revised manuscript.

“These findings of this study echoed the SOM experiences identified in previous studies conducted in the North America and Hong Kong such as endorsement of heteronormative culture and behaviors, assumption of psychological abnormality, and avoiding contacts due to homophobia [6,1214,17,2931]. Please refer to line 376-380.

“The present study revealed that sexuality stereotyping is a common type of microinsult. Similar SOM manifested with stereotypical assumptions were also identified in previous studies conducted in the North America [6,12,13,29,31].Please refer to line 395-397.

The notion of clear-cut gender roles that conform to heterosexual relationships is another common sexuality-related stereotype. This SOM rooted in the recognition that the standards of heterosexual individuals are the only norm in the world [6,1214,17,29,30]. LGB individuals may face the impolite inquiries from heterosexist acquaintances or strangers. However, the perpetrators of microinsults might underestimate the harm that they cause to LGB individuals. Similarly, heterosexual individuals often deny any biases or unfavorable attitudes toward LGB individuals they may hold [6]. Please refer to line 400-406.

“The present study revealed that people often perceive same-sex relationships as a trend and invalidate their existence. Previous studies have also revealed that questioning the reality or importance of LGB individuals’ feeling and experiences is one of common SOMs in western societies [14,30,31].” Please refer to line 415-418.

Reviewer 2 Report

Dear authors,

It was a pleasure to review this manuscript that attempts to study the sexual orientation microaggression experiences and coping responses of lesbians, gay men, and bisexuals in Taiwan through a qualitative study.

I found the topic extremely interesting since where heteronomativity prevails, attacks on minority groups can even be accepted.

With the sole objective of being able to improve the quality of the manuscript, I would like to make some comments:

It is not clear from the methodology how the sample was chosen. The authors say that 50 people were invited and only 30 of them agreed to participate. My question is why did the authors settle for 30 participants? Did the 3rd reach data saturation?

Another weakness that I see in the methodology is that it does not clearly explain how the information was processed. Was any statistical software used in the qualitative studies? How was the information coded and what study was done to see the consistency of the data? I think this should be better explained in the study.

For the rest, the article seems acceptable to me.

Kind regards

Author Response

We appreciated your valuable comments. As discussed below, we have revised our manuscript with underlines based on your suggestions. Please let us know if we need to provide anything else regarding this revision.

Comment 1

It is not clear from the methodology how the sample was chosen. The authors say that 50 people were invited and only 30 of them agreed to participate. My question is why did the authors settle for 30 participants? Did the 3rd reach data saturation?

Response

Thank you for your comment. In the revision, we explained that originally this sample size was proposed when we applied for a research funding for this study according to Guest et al.’s advice. We eventually settled on this sample size as the interview data appeared adequate to answer the research questions. Please refer to line 144-150.

The present study sent the invitation message by cell phones to the 50 LGB individuals who obtained the highest SOMI scores on the T-SSS to participate in qualitative interviews. In total, 30 LGB individuals (17 women and 13 men) responded to the invitation and agreed to participate. While we estimated this sample size to be enough according to Guest et al.’s suggestion when we composed a funding proposal for this study, interview data from this number of participants appeared adequate to answer the research questions [39].

Comment 2

Another weakness that I see in the methodology is that it does not clearly explain how the information was processed. Was any statistical software used in the qualitative studies? How was the information coded and what study was done to see the consistency of the data? I think this should be better explained in the study.

Response

We added the explanations for how the information was processed below.

Two researchers (YTH and YCH) jointly performed a thematic analysis with the aid of NVivo 10 software to analyze the transcripts [40]. Content analysis was not applied since the study objective was to offer comprehensive descriptions of SOM experiences rather than to determine the prevalence or intensity of a certain form of SOMs. At the first stage, two coders read the transcripts thoroughly to familiarize themselves with the data. Stage Two involved initial line-by-line coding, where descriptive codes were assigned to indicate the meanings in the text. At Stage Three, the analysts conducted focused coding to identify themes from the codes according to their nature and relevance with experiences and coping methods of microaggression.” Please refer to line 174-182.

The coded themes were then reviewed through regular research team discussion to assess whether the themes had been consistently and properly developed.” Please refer to line 189-190.